# Actomyosin Contractility in the Generation and Plasticity of Axons and Dendritic Spines

**DOI:** 10.3390/cells9092006

**Published:** 2020-09-01

**Authors:** Marina Mikhaylova, Jakob Rentsch, Helge Ewers

**Affiliations:** 1RG Optobiology, Institute of Biology, Humboldt Universität zu Berlin, 10115 Berlin, Germany; 2DFG Emmy Noether Group ‘Neuronal Protein Transport’, Center for Molecular Neurobiology, ZMNH, University Medical Center Hamburg-Eppendorf, 20251 Hamburg, Germany; 3Institut für Chemie und Biochemie, Freie Universität Berlin, 14195 Berlin, Germany; jakob.rentsch@fu-berlin.de

**Keywords:** actin, myosin, spectrin, calpain, calcium signaling, AIS, dendritic spines

## Abstract

Actin and non-muscle myosins have long been known to play important roles in growth cone steering and neurite outgrowth. More recently, novel functions for non-muscle myosin have been described in axons and dendritic spines. Consequently, possible roles of actomyosin contraction in organizing and maintaining structural properties of dendritic spines, the size and location of axon initial segment and axonal diameter are emerging research topics. In this review, we aim to summarize recent findings involving myosin localization and function in these compartments and to discuss possible roles for actomyosin in their function and the signaling pathways that control them.

## 1. Introduction

Neurons are highly specialized cells with an exceptional degree of spatial compartmentalization. Despite of a large morphological and functional diversity of cell types, most neurons possess long, thin processes known as axons and branched dendrites that can extend for distances several orders of magnitude higher than the size of the cell body they emanate from. At the same time, this extreme shape can persist for decades virtually unchanged. Clearly, axons, which are thousands of times longer than they are in diameter, experience great mechanical stress. They must be sufficiently stiff to resist mechanical tensions and not tear, but remain flexible enough to accommodate for structural plasticity that may be required for their functional adaptability. Such mechanical and structural properties are generated and maintained by the cytoskeleton in conjunction with force-generating molecular motors. An especially prominent role is played here by actomyosin, a network of interconnected actin filament bundles that are pulled together by myosin motors, especially non-muscle myosin 2 (NMII, Box 1). Actin and NMII are evolutionarily old molecules and both are ubiquitously expressed, including the vertebrate central nervous system. NMII has a well-described role in neurite elongation, axonal outgrowth and neuronal polarization, as is abundantly present in neuronal growth cones, where it controls microtubule bundling and regulates the actin-rich lamellipodium and filopodia [1,2,3,4]. More recently, NMII has been localized to dendritic spines and the axonal initial segment (AIS) [5,6,7,8,9,10]. The topic of this review is actomyosin functions in these important neuronal subdomains which in contrast to the growth cone are much less well understood. 

Box 1Actomyosin in the nervous system.The term actomyosin generally describes contractile bundles assembled of actin filaments that are interconnected by bipolar bundles of myosin II. The motor domain of myosin II can execute a power stroke after hydrolysis of ATP and release of phosphate, moving the myosin molecule relative to the actin cable towards its barbed end in the process. This pulls the actin filaments on both ends of the myosin bundle closer together, leading to contraction of the actomyosin structure. Binding of a new ATP molecule leads to unbinding of the motor domain from actin and new binding further upstream the actin filament for progression of the movement. This mechanism of contraction is the basis for muscle function and many mechanical processes in cells. Generally, myosin II isoforms for the skeletal, cardiac and smooth muscle, which are specialized to function in elongated thick bundles called sarcomers, are distinguished from three non-muscle myosin II (NMII) isoforms that execute many different mechanical functions in cell biology. NMII exists as a hexamer that consists of two copies each of elongated heavy chains bearing motor domains, two regulatory light chains and two essential light chains that stabilize the heavy chain structure. These hexameric units with two motors on one end and an elongated coiled coil on the other end further bundle both in a parallel and antiparallel manner into bipolar structures that can pull actin filaments together. The essential light chain stabilizes the hexamer and phosphorylation of the regulatory light chain at serine 19 is required for NMII to be able to execute its power stroke. The three NMII isoforms NMIIA, -B and -C are encoded by the heavy chain genes MYH9, MYH10 and MYH14, respectively. All these isoforms are expressed in most non-muscle cells and NMII A and B are expressed highly in the nervous system. There is no detectable difference in actin binding or activation between non-muscle myosin II A, B and C [11], but the isoforms differ in subcellular localization and some biochemical properties such as ATPase activity and duty ratio. Most mammals have six actin genes, four of which are expressed mostly in muscle. The cytoplasmic β-actin (ACTB), a complex locus with 22 introns and 23 splice isoforms, and γ-actin (ACTG1), with 16 introns and 19 mRNAs that encode for 15 different isoforms in a highly tissue-specific manner, are both expressed in the nervous system. In this review, when we use the term actomyosin, we refer to bundles and networks of NMII and β-actin or γ-actin.

## 2. Neuronal Morphology and Compartmentalization

Most neurons in the central nervous system undergo a defined developmental program that starts with the growth of several processes from the cell body or soma. The microtubule cytoskeleton plays an essential role in providing structural support for growing neurites, whereas a dynamic, branched actin cytoskeleton enriched at their tips in so-called growth cones is important for giving the directionality and further differentiation of the neuron. One of these processes poised to become the axon then undergoes a period of quick continuous growth that requires the generation of bundles of microtubules that are generated through de novo polymerization and microtubule transport [12], and the activity of cdc42 [13,14]. As a result, the neuron is polarized into somato-dendritic and axonal compartments. After a period of axonal outgrowth, the dendrites start to develop more and more complex branches and form hundreds of contact sites with axons from other cells. Stabilization of these connections between neurons and the recruitment of post-synaptic components, as well as pre-synaptic vesicles and secretion machinery, leads to synapse formation and specialization in the membrane composition of the pre- and post-synaptic sites. In mature neurons the majority of excitatory post-synapses are located to the flattened tips of bulbous protrusions called dendritic spines, where ion channels, receptors and adhesion molecules supported by scaffolding proteins are enriched in a membrane domain called the post-synaptic density (PSD, Box 2). This PSD is organized in nanodomains and tightly apposed across the synaptic cleft to a corresponding synaptic vesicle release machinery in the pre-synapse [15,16]. Dendritic spines can vary in their morphology, size and molecular composition, and the formation, plasticity and stability of the synapse bearing spines is thought to be the physical correlate to learning and memory formation. The shape and geometry of neurons are thus closely connected to their function in the size range of neuronal compartments varies from elongated axons that can span many centimeters to deliver action potentials to the low micrometer scale of dendritic spines. Their highly branched and elongated processes reflect the network architecture of the nervous system and both the tubular narrow shape of the axon as well as the thin neck of the dendritic spine are thought to play important biophysical roles in regulating neuronal electrical signaling via the ion flow and in creating biochemically compartmentalized subcellular domains. The spatial and geometric organization of signaling molecules and structures is thus tightly regulated, and these functions are executed by the cytoskeleton.

Box 2Dendritic spines.Dendritic spines are small protrusions of the dendritic shaft that are the postsynaptic site of excitatory glutamatergic synapses, which comprise a pre-synaptic terminal or bouton separated by the synaptic cleft from a specialized membrane domain called the post-synaptic density (PSD). While the pre-synaptic site contains numerous neurotransmitter vesicles spatially arranged for swift membrane fusion by proteins forming a cytomatrix of the active zone and highly sensitive vesicle release machineries, the PSD accommodates different types of glutamate receptors and calcium channels anchored to scaffolding proteins. Calcium signaling plays an essential role in evoking vesicle release upon neuronal depolarization at the pre-synapse as well as in triggering calcium-dependent kinase and phosphatase pathways, such as CaMKII or calcineurin at the PSD. Calcium signaling via the calcium binding proteins calmodulin and caldendrin is transduced to various actin modifiers, thereby directly regulating the morphology of dendritic spines in response to stimuli. Pre- and post-synaptic sites are tightly connected via cell adhesion molecules forming hetero- and homophylic interactions between each other. The most common type of dendritic spines in the adult brain are mushroom-like spines. Their shape, with a bulbous head and thin neck, is important for the compartmentalization of synaptic signaling and provides input specificity to this very synapse. The specific shape of dendritic spines and their ability to undergo rapid changes or a long-term stabilization of their structure to a large degree depends on the actin cytoskeleton. Thousands of dendritic spines can be found on branched dendrites and excitatory synaptic plasticity is accompanied by changes in a number of AMPA receptors and in the head size of dendritic spines that are thought to be the physical correlate of learning and memory formation.

While the entire cytoskeleton has important roles in neurons, the role of neurofilaments and microtubules have been extensively reviewed in [14,17]. Here we will focus on actomyosin cytoskeleton in narrow neuronal compartments, whereas its role in the growth cone has been discussed elsewhere [18,19]. We will summarize recent discoveries on the structure and components of the neuronal actomyosin network and discuss mechanisms involved in regulation of contractility and its relationship to neuronal functions, such us plasticity of the axonal initial segment (AIS) or dendritic spines. 

## 3. The Membrane-Associated Periodic Skeleton (MPS) in Neurons

In one of the first breakthrough discoveries of single molecule super-resolution microscopy (SMLM), a periodic arrangement of ~200 nm spaced actin rings has been found along the axon that is interconnected by bipolar spectrin tetramers [20]. This finding came unexpected, as this striking structure had never been observed in decades of investigation of axons with electron microscopy, which is most likely due the very frail nature of actin filaments that easily get disrupted upon a treatment with detergents. Since then, the existence of a periodic MPS has been seen by a number of microscopy methods and in several laboratories [5,21,22,23,24,25]. Importantly, the nanoscopic organization of the components of the MPS has recently been verified in platinum replica electron microscopy of unroofed cultured neurons [26]. Strikingly, a new organization of actin filaments into ~1 µm long braids of two actin filaments has been found here. The MPS structure seems to be conserved from worm to humans, and by now it has been found in a variety of excitatory and inhibitory neurons of the central nervous system and motor neurons of the peripheral nervous system [27,28]. While the MPS was initially only observed in neuronal axons, it has since been observed in neuronal stem cells and shown to be present in astrocytes and oligodendrocytes as well [29]. It is also found in dendrites [23,30], specifically at the neck of dendritic spines [24,31].

The MPS emerges early in neuronal development and can be detected from two days in vitro (DIV) onward progressing from the proximal axon, where it precedes all axon initial segment makers [20]. Braided double actin filaments of the MPS align the cylindric axon along its entire length and colocalize with adducin, tropomyosin and the phosphorylated form of the myosin light chain. These ring structures are interconnected by spectrin tetramers consisting of two α and two β spectrins, with the N-termini of the β spectrins connected to actin and the C-termini located in the center between actin rings [26]. Spectrins are required for MPS assembly [30] and the mechanical stability of axons [32]. Indeed, in spectrin knockout animals, axons can tear in response to mechanical stress induced by movement of the animal [32], suggesting a structural role of the MPS in axonal stability [33]. The β spectrin isoform in the MPS spectrin tetramer seems to vary with the specific location in the cell with spectrin βIV in the AIS, spectrin βII in the distal axon and dendrites and βIII largely in dendrites [16,26,29] (Figure 1). Although the periodic pattern of βIII spectrin in spines has not been directly confirmed, only a fraction of dendritic spine necks contains βII spectrin, suggesting that other spectrin isoforms might be responsible for spacing of the actin rings at this site (Figure 1). Indeed, spectrin βIII has been found at the base of dendritic spines and shRNA knock down of this isoform results in collapse of dendritic spines into shaft synapses [34]. Together, these findings suggest that the neuronal MPS may exhibit locally distinct compositions and β spectrin isoforms may fine-tune its structural properties. 

A special type of the MPS is present in the AIS which is a stretch of 50–100 µm at the beginning of the axon that separates the axonal and somatodendritic domains of neuronal cells [35,36]. The AIS contains a specific complement of cytoskeletal and adaptor proteins that cluster adhesion proteins and several types of ion-channels, which are responsible for the generation of action potentials. The most prominent AIS-specific molecules are spectrin βIV and a large AnkyrinG (AnkG) isoform, which directly interact, and the cell adhesion protein neurofascin. These molecules are organized in an MPS of higher complexity than it is found in the distal axon. In analogy to axonal spectrin βII, in the AIS, spectrin βIV interconnects braided actin rings in a ~200 nm periodicity. AnkyrinG binds to spectrin βIV between the actin rings and this scaffold recruits neurofascin and ion channels in a manner that is evolutionarily conserved from vertebrates onward (Figure 1). Such high local density of ion channels allows for action potential initiation. In addition to this, AnkyrinG and the AIS are required for the maintenance of neuronal polarity [37,38]. Specifically, if AnkG is downregulated by shRNA in cultured cells or in vivo, axons lose their molecular identity and start to acquire dendritic features including postsynaptic densities that form adjacent to nearby presynaptic terminals of other axons [38]. In a situation of hypoxia, such as in ischemia, the AIS breaks down [39], and consequently, neuronal polarity is lost, resulting in severe pathological consequences [39]. This process is mediated by the calcium-dependent cysteine protease calpain [39]. Even though the AIS persists for the entire lifetime of a neuronal cell, it is remarkably plastic and dynamic. First, as described above, it assembles in a dynamic process during the first week of neuronal development. Secondly, in response to chronic depolarization, the entire AIS structure can change in length and shift in its location along the axon, resulting in modified action potential generation [40,41]. Recently, a prominent role of non-muscle myosin II and regulating molecules in the plasticity of the AIS and the MPS in response to Ca2+ signaling as discussed below has emerged [6,42].

## 4. Regulation of the MPS

The unique organization of the MPS is perfectly suited to provide structural durability and resistance to mechanical forces that axons or dendritic spines encounter during intracellular trafficking, remodeling of neuronal morphology or a tissue movement. Indeed, axons break more readily when spectrin is absent [32]. Filamentous actin and spectrins are the most important structural elements of the MPS as pharmacological studies using actin-depolymerizing drugs such as cytochalasin D and latrunculin or down-regulation of spectrin’s expression resulted in disintegration of the MPS. Interestingly, the MPS is more resistant to actin depolymerizers than actin filaments at dendritic spines, filopodia or axonal actin cables [22]. Such resistance is increasing with neuronal maturation and correlates with developmental expression profile of the actin-capping protein α-adducin [43]. Moreover, there are location-specific differences in MPS stability with the AIS being particularly resistant to high doses of cytochalasin D and latrunculin, suggesting that there might be further differences indicating further diversity in the composition of the membrane cytoskeleton [22]. 

Earlier experimental and modeling data suggested that the flexibility of the MPS along the axon might be limited because the spectrin filaments are already held under entropic tension [44]. Therefore, for example, injury-induced disruption of the cortical axonal cytoskeleton cannot be spontaneously restored [44]. However, the axon is an elastic compartment that can undergo radial deformations for instance during trafficking of a large cargo or neuronal activity [5,45]. The length and the localization of the AIS are changed during neuronal development and as response to neuronal activity which would require lateral assembly/disassembly-based movement of its specialized MPS. Indeed, recent discoveries indicate that the MPS can undergo reorganization upon very specific types of stimuli and the role of calcium signaling and the actomyosin network are currently the central focus [42,46]. 

### 4.1. Regulation of the MPS by Calcium

The AIS disappears in brain regions affected by ischemic injury and oxygen-glucose depletion, which lead to degradation of βIV-spectrin and AnkG by the calcium-dependent cysteine protease calpain [39]. MPS βII-spectrin can likewise be digested by calpain-2 upon activation of ERK signaling [46]. Calpain-1 and -2 are major cellular proteases that are critical for proper neuronal branching and dendritic spine complexity [47]. Increased calcium concentrations activate calpains and are known to lead to the cleavage of several major cytoskeletal proteins, such as MAP2, spectrin, vimentin, internexin and others [48,49,50]. Therefore, it is not surprising that calpains play an important role in cytoskeletal reorganization in neurons as well. The axonal MPS serves as a signaling platform for receptor tyrosin kinases (RTK), adhesion molecules and G-protein coupled receptors, which are corralled by the periodic actin rings [51] and immobilized by them in response to extracellular stimuli [46]. Functional cross-talk between clustered receptors triggers the activation of ERK kinase signaling, which in turn activates calpain-2, resulting in a rapid MPS degradation. Removal of the constraints, limiting the diffusion of membrane proteins, contributes to the termination of RTK trans-activation signaling. It is possible that similar mechanisms of MPS re-organization will be present not only in the axon but in dendrites and the neck of dendritic spines. It is well-known that synaptic firing results in a calcium influx which triggers activation of calpains [52] and the disruption of the dendritic MPS by neuronal activity was found to be dependent of NMDAR-mediated calcium influx [53]. Calpain-2 is also known as m-calpain because of its millimolar calcium binding affinity in vitro [54]. Since intracellular calcium concentration rarely exceeds 1 µM (resting state below 100 nM, activated state below 1 µM [55], maximal synaptic concentration 50 µM [56]) at physiological conditions, it is possible that in vivo calpain-2 is associated with other proteins, which make it more sensitive to calcium. Accordingly, calpain-2 is implicated in a number of pathological states, including neurodegenerative disorders like Alzheimer’s disease [57,58]. Therefore, future studies should address to what extend calpain-2 contributes to the MPS remodeling in different neuronal compartment, what kind of stimuli can trigger calpain-2-dependent degradation of MPS, how high the intracellular calcium concentration should be and how local this process is. It seems unlikely that the axonal RTK signaling will be triggered along extended membrane areas in vivo since even few hours of reduced axonal stability could have detrimental consequences for neuronal function and survival. 

### 4.2. Non-Muscle Myosins in Regulation of the MPS

Non-muscle myosin II motor proteins (represented by myosin IIA, myosin IIB, and myosin IIC) have recently attracted strong interest in respect of MPS regulation. Functional myosin II motor protein complex consists of two myosin heavy chains (MHCs), two essential light chains (ELCs) and two regulatory light chains (MLCs, see Box 1). In contrast to myosin V and VI, which are processive myosins involved in cargo trafficking along actin filaments, myosin II family members are contractile myosins assembled into antiparallel bundles that pull actin filaments together. Powered by ATP, they participate in a multitude of neuronal processes including reorganization of actin filaments in the axonal growth cone dynamics, structural plasticity of dendritic spines and the AIS, signaling to RhoGTPases and many others. Structurally, the MHC is subdivided into three distinct regions called motor, neck and tail domains. The globular motor domains transduce the energy released during hydrolysis of ATP into active mechanical force and processive movement along actin filament. The neck region connects the motor domain to the elongated coiled coil that forms the tail domain and has an important regulatory role. The two conserved IQ motifs in the neck domain control NMII switching between a folded (inactive) or open (active) conformation. These IQ motifs can bind calmodulin, but in the case of NMII they are usually occupied by ELC and MLC [59] Binding of the ELC to the neck region provides stability to the neck domain of the MHC whereas MLC binding also allows regulation of motor activity through phosphorylation at two subsequent threonine and serine residues (T18/S19) [60]. The tail region coiled-coil assembles two NMII molecules into the constitutive homodimer. Activated myosin II dimers form an elongated shape and such hexameric complexes can bundle in a parallel and antiparallel way into bipolar filaments of approximately 300 nm in length [61]. The myosin II motor domains are flanking both sides of the filament and their stepping motion in the barbed-end direction of oppositely oriented actin filaments results in actomyosin contraction. The structure and regulation of myosin II filaments in muscle and non-muscle cells have been recently reviewed by Dasbiswas et al. [62]. Interestingly, in neurons actomyosin together with the MPS are implicated in regulation of the AIS and axonal diameter and action potential firing [5,45]. 

It has been known already for some time that repeated neuronal depolarization can lead to a reversible repositioning and length change of the AIS, which is crucial for homeostatic control and plasticity of neuronal excitability [41,63]. This type of AIS structural plasticity occurs within a few hours after stimulation and involves AIS disassembly in a more proximal part resulting in the AIS shortening following more long-lasting extension towards the distal part of the axon [40,63]. Myosin II was implicated both in rapid disassembly as well as relocation and extension, since the selective non-muscle myosin II inhibitor blebbistatin completely blocked any activity-dependent morphological alterations [42]. In recent work, Berger et al. provided deeper insight into the mechanisms involved in AIS relocation and identified actomyosin as a key element [6]. Myosin light chain (MLC) can activate myosin II when it is mono- or di-phosphorylated at S18 and Ser19 residues [64]. The di-phosphorylated form of the MLC (ppMLC) is highly enriched in the AIS, where it associates with the actin rings of the MPS [6]. MLC dually phosphorylated in this way stimulates myosin II contractility, which in turn is required for AIS assembly and the faithful distribution of AIS components. During neuronal depolarization, ppMLC is rapidly lost, which results in the destabilization of actin filaments, which in turn allows for a remodeling of the AIS [6,42]. Interestingly, ppMLC was hardly detected in other axonal compartments and dendrites, suggesting that this mechanism of regulation is very specific for the AIS. Myosin light chain phosphorylation and AIS relocation both are calcium dependent, as calcium chelation and the inhibition of L-type calcium channels completely prevented loss of ppMLC and AIS disassembly [6]. One calcium signaling pathway underlying this process seems to involve calpains, as the calpain-1 and -2 inhibitor MDL28170 partially rescued ppMLC levels and prevented AnkG loss. Furthermore, calcineurin is a calcium- and calmodulin-dependent serine/threonine protein phosphatase, which was implicated in the regulation of the AIS relocation upon neuronal depolarization [40,42]. Indeed, the calcineurin inhibitor CsA reduced the levels of ppMLC and AnkG to varying degrees under resting conditions [6]. This suggests that various calcium signaling cascades converge in the regulation of different steps of the AIS remodeling upon induction of plasticity. 

The role of actomyosin in the neuronal cytoskeleton goes beyond regulation of the AIS plasticity. Two very recent studies reported that the axonal MPS is a contractive actomyosin network, which facilitates structural stability of the axon, regulates axon diameter and radial contraction as well as cargo trafficking and axonal propagation of electrical signals [5,45]. Thin axons with an inner diameter of less than 1 µm are very abundant in the mammalian central nervous system [65]. This poses obvious challenge for the transport of large cargoes, such as autophagosomes, mitochondria, endosomes or lysosomes, as their diameter can exceed the diameter of the axon [45]. Using live super-resolution imaging in combination with pharmacological approaches and electron microscopy, it could be shown that the passage of large axonal cargoes causes a transient radial expansion of the axon followed by constriction which depended on myosin II activity [45]. A short-term inhibition of myosin II activity did not change the periodicity of actin rings, but led to an increase in axon diameter instead, suggesting that myosin II activity keeps the MPS under constant tension to control ring size. SIM imaging revealed that the motor domain of myosin II but not the tail-domain involved in dimerization colocalized with the periodic, braided actin rings [45]. 

Work from the Sousa laboratory provided further molecular insights on the regulation of myosin II contractility along the axon. Similar to the AIS, they found that phosphorylation of the MLC plays a critical role in the contractility of the axonal actomyosin network. Of note, they used pharmacological inhibition of MLC phosphorylation and antibodies detecting di-phosphorylated S18/19 MLC, which cannot distinguish between either mono- or the di-phosphorylated forms of MLC. It remains therefore unclear, whether different states of phosphorylation lead to different outcomes. What is clear is that MLC phorphorylation by myosin light chain kinase triggered conformational changes and self-assembly of myosin II complexes in filaments, leading to a constriction of the axonal diameter. It was found that changes in axonal width did not affect the angle of actin rings in relation to the axonal axis. This suggests that that myosin II probably does not contract between adjacent rings [5].

Another example of narrowed compartments are dendritic spines where the thin spine neck also contains an MPS [24,31]. NMII has been found in dendritic spines [7], and is found in about 90% of dendritic filopodia [66]. It localizes mostly in the neck and head region [66]. In the neck region, myosin II molecules were found to be organized in linear clusters, suggesting the assembly of myosin bundles, whereas in the head, single molecules were found in immunogold EM [66]. NMII is required for shortening of the spine stalk in spine maturation and spine development [8,67] in a process that involves RLC phosphorylation via Rho-kinase (ROCK). NMII activity is required both for NMDAR-dependent LTP and for the LTP-dependent spine actin polymerization required for structural plasticity of dendritic spines [68] as evident from knockout animals. These effects are dependent of MLC phosphorylation, which is controlled by NMDAR signaling. Indeed, myosin is required for memory formation and consolidation [68]. For a detailed review of synaptic myosins, see [69]. Unlike in the AIS, a role for NMII in the organization of the MPS at the neck of dendritic spines has not been yet investigated.

## 5. Perspectives 

Since the initial discovery of the periodic membrane cytoskeleton in axons [20], the molecular composition of the MPS [20], its distribution over neuronal compartments [23,24,30,31], its appearance during development [43] as well as its presence across different cell types cells of the nervous system [28,29] and its evolutionary conservation from worm to mammal [27] has been described in quick succession. In a study using the nematode Caenorhabditis *elegans* as a model, it has been shown that the MPS plays an essential role during tissue movement: it provides mechanical support and elasticity to the axon [70]. However, only very recently the regulation of MPS assembly and disassembly and its cellular functions have begun to emerge. In this respect the degradation of the MPS via calcium/calpain-2 is an efficient and elegant mechanism how the neuron could locally reorganize its membrane cytoskeleton and terminate RTK signaling in axons [46]. Fundamental questions remain unanswered. It remains unclear whether this is an axon-specific pathway or whether the MPS in other neuronal compartments, such as the AIS, dendrites or the neck of dendritic spines, may undergo a similar type of regulation. A possible tuning of MPS sensitivity to degradation could stem from differences in susceptibility to calpain-cleavage of different β-spectrin isoforms. The availability of calpain-2 and the possible contribution of other calpain proteases could be additional factors in the local modification of MPS degradation. It is still an open question how the MPS can recover. It will be important to understand what proteins and signaling pathways can catalyze its assembly and what forces can bring spectrin tetramers into the energetically unfavorable extended arrangement [44] in order to bridge neighboring actin rings. 

A new exciting aspect of the biology of the MPS is its inclusion into a contractile actomyosin network. This recent finding offers an explanation as to how narrow cellular processes could dilate to allow the passage of large cargo or could scale their diameter in response to neuronal activity [5,45]. The exact spatial relationship between the actin rings and the position of myosin II motor complexes remains unclear. Taking into account that the length of an active two-headed myosin motor complex is around 300 nm and that the distance between actin rings is 190 nm, actin rings may connect to myosin head groups in different ways. First, the myosin II bundle could crosslink two neighboring rings with an angle deviant from 90°. This could be achieved if myosin motors connected to neighboring actin rings in a one-dimensional lattice (similar to spectrin) or across the axonal volume (Figure 2). In such a configuration, the force generated by myosin II would induce filament sliding within the braid that would results in constriction or expansion of the ring. However, as myosin II steps towards the barbed end, the orientation of filaments within one braid and filament polarity with respect to the neighboring rings will be of critical importance. Such a mechanism can work if (i) two filaments within the same braid have an opposite polarity, (ii) when filaments assembling the same braid are parallel but the adjacent braids have opposite polarity, or (iii) when the rings are composed of not one but several non-uniformly oriented braids and there is at least one filament present in the opposite orientation in two neighboring rings. In an alternative scenario, myosin bundles could cross inside a ring in a radial manner. For the myosin bundle to change ring diameter, actin filaments with opposite polarity would be required to allow for ring extension and contraction (Figure 2). A sarcomere-like organization of neuronal MPS is also unlikely because this would imply that multiple myosins are organized in a filament with motor domains pointing outside. The diameter of such filament is approximately 30 nm, which is much less than the spacing between actin rings. Of note, a replica EM also did not reveal such myosin structures in the axon [26]. When considering the radial myosin organization model, it should be kept in mind that the cytosolic content of the axon, especially microtubules, intermediate filaments and the endoplasmic reticulum, can become a physical obstacle that restricts a number of myosin II complexes linked to the MPS. Cryo-electron microcopy could help in addressing some of the questions. For instance, what is the orientation of individual actin filaments in a braid; are all braids built uniformly and how are they made? Perhaps it would be possible to visualize the myosin II complex together with the MPS. Understanding the mechanisms of braid assembly will shed further light on their regulation and resistance to depolymerization. One can speculate that in the analogy with microtubules [71], actin braids can be made by the capability of myosin II to cross-link and slide the actin filaments. This could cause helical motion of overlapping filaments around each other and thus allow formation of actin braids. Arrangement of filaments into a helix could influence a pitch angle of the individual filaments and thus change their affinity for cofilin binding, thus making them more resistant to a severing. As incorporation of different actin isoforms and the post-translational modifications can also change the stability of actin filaments, it remains to be tested whether some of those are particularly enriched at the MPS [11,72]. Furthermore, actin braids are a very unusual type of filament organization. It will be very interesting to reproduce and explore biophysical properties of such filaments in in vitro reconstitutes systems.

Actomyosin and organelle trafficking in narrow compartments, such as axons of the neck of dendritic spines, is another very interesting area of research that has recently emerged. Future studies using improved live super-resolution microscopy techniques could highlight the spatio-temporal kinetics of actomyosin response. Research in this direction is warranted as it might be applied to improving transport properties in axons, which can be relevant for aggregation clearance, which in turn might be beneficial in a number of neurodegenerative disorders, such as Parkinson’s disease or Alzheimer’s disease.

## Figures and Tables

**Figure 1 cells-09-02006-f001:**
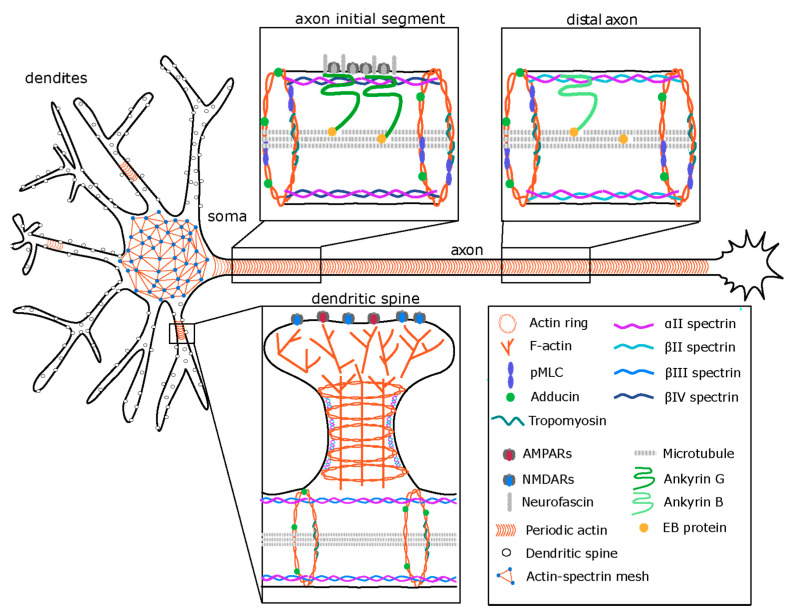
The organization of the neuronal cytoskeleton. A two-dimensional (2D) actin-spectrin meshwork, similar to those found in other cell types (e.g., erythrocytes), spans the soma of the cell. In contrast, a one-dimensional (1D) periodic membrane cytoskeleton (MPS) is found in axons, in a fraction of dendrites and at the neck of dendritic spines. Top: The MPS consists of actin rings at a periodicity of ~200 nm, interspersed with spectrin tetramers. Each actin ring is formed by two braided actin filaments. The actin rings are further stabilized and regulated by the capping protein adducin and by tropomyosin. The spectrin tetramers are comprised of two αII spectrins and two compartment specific isoforms of β-spectrin. The axon initial segment (AIS) is a stretch of 50–100 µm at the beginning of the axon. The major scaffold in the AIS is AnkyrinG (AnkG), which binds to spectrin βIV and recruits the adhesion molecule neurofascin and ion channels. Phosphorylated myosin light chain (pMLC) is localized to actin rings in the axon. Microtubule bundles are stabilized by plus end binding proteins (EB) along the axon in both the AIS and the distal axon. In the distal axon the MPS is organized by AnkB, which in turn binds to βII-spectrin. AnkB is also arranged periodically, though the pattern is less prominent. Bottom: Dendritic spines are important for compartmentalization of synaptic signaling conferred by glutamate receptors and calcium channels. While the head of the spine contains branched actin filaments, the MPS is prominent in the neck region. Likely consisting of acting rings interspersed by αII and βII-spectrin tetramers. The MPS has also been observed in a sub-fraction of mature dendrites. Here, spectrin tetramers contain the βIII-spectrin isoform.

**Figure 2 cells-09-02006-f002:**
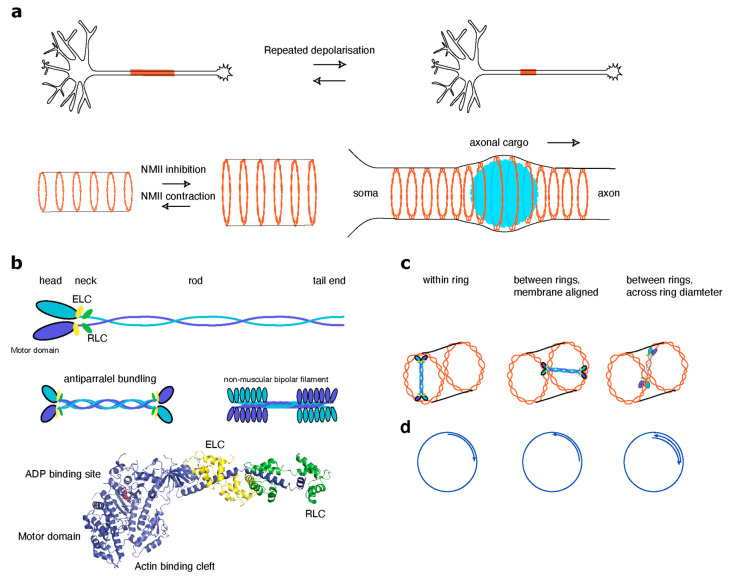
Organization and function of non-muscle myosin II (NMII). (**a**, top) Repeated depolarization leads to proximal shortening of the AIS and subsequent extension towards the distal axon. NMII activity is necessary for this process as blebbistatin completely blocks the activity. (**a**, bottom) NMII activity controls the axonal diameter. Inhibition of NMII by blebbistatin leads to an increase in axon diameter, indicating that NMII holds the membrane-associated cytoskeleton (MSK) under constant tension. In addition, NMII has been shown to be implicated in cargo trafficking along the AIS. The size of large cargo (e.g., autophagosomes, mitochondria, endosomes or lysosomes) can exceed the diameter of the axon. Passage of this large axonal cargo causes a transient radial expansion of the axon followed by constriction, which depend on myosin II activity. (**b**, top) Schematic organization of NMII. NMII exists as a hexamer that consists of two copies each of elongated heavy chains, two regulatory light chains (RLC) and two essential light chains (ELC) that stabilize the heavy chain structure. The heavy chain is composed of an N-terminal motor domain, a neck domain, which interacts with both light chains, an α-helical rod domain and a C-terminal tail. (**b**, middle) The hexameric units further bundle both in a parallel and antiparallel manner into bipolar structures that can pull actin filaments together. (**b**, bottom) Crystal structure of the motor and neck domains of NMII interacting with ELC and RLC. The motor domain contains the actin binding cleft where NMII interacts with actin. Shown in red is ADP bound at the nucleotide binding site. Cycling from ATP to ADP at the nucleotide binding site leads to conformational changes in the actin binding cleft, which modulate the interaction of NMII with actin. (**c**) Models of a possible spatial relationships between NMII and actin rings. The length of an active two-headed myosin motor complex is around 300 nm, while the distance between actin rings is only 190 nm. (**c**, left) NMII crosses the diameter of a single actin ring. Alternatively, NMII could cross link two neighboring rings with an angle deviant from 90°, which can be achieved when myosin motors connect neighboring actin rings in a one-dimensional lattice (as spectrin) (**c**, middle) or across the axonal volume (**c**, right). (**d**) Polarity of actin filaments in ring-forming actin braids. Force generated by NMII induces filament sliding within the braid that results in constriction or expansion of the ring. However, as myosin II steps towards the barbed end of an actin filament, the orientation of filaments within a single braid and filament polarity with respect to the neighboring rings are important. The NMII mechanism can work when filaments within a single ring are parallel (**d**, left) but the adjacent braids have opposite polarity (**d**, middle) or when two filaments within the same braid have an opposite polarity (**c**, right).

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
