# Peer review of "Actomyosin Contractility in the Generation and Plasticity of Axons and Dendritic Spines"

_cells, 2020, doi:10.3390/cells9092006_

Round 1

Reviewer 1 Report

The Review focuses on the role played by actin and non-muscle myosins in the development and homeostasis of axons, dendrites, and dendritic spines.

The authors give a lot of information and are fairly detailed, however, the role of myosins gets somehow lost.

Maybe if the authors split the chapters

Chapter 4 for example could be divided into two, a first part “Regulation of MPS” and an additional chapter more focused on non-muscle myosins in the MPS.

The “Perspectives” chapter is far too long, and a bit convoluted as a final perspective chapter.

The figures are very well done, I like it a lot.

Author Response

Please find our response in the attached file

Reviewer 2 Report

The authors present a focused review of the role of the submembranous cytoskeleton of neurons with emphasis on recent studies addressing the role of actomyosin contractility along axons and the axon initial segment (AIS), and in dendritic spines. The manuscript discusses the relevant literature and is up to date. However, the manuscript would benefit from reorganization, comprehensive editing and corrections. There are instances wherein the references do not match with the statements they are presumably supporting.

Title of the manuscript:

“Actomyosin contractility in the generation and plasticity of neuronal compartments”.

Considering that actomyosin contractility has many roles in many subcellular compartments at different stages of neuronal development, the title is considered rather broad for a review that focuses on two select subcellular compartments (the axon and the AIS, and dendritic spines). Based on the title this reader was expecting a much more comprehensive review of the field. It is suggested the title be reconsidered to address the specific compartments addressed by the review; e.g., “Actomyosin contractility in the generation and plasticity of axons and dendritic spines”.

Organization of the manuscript:

The manuscript focuses on the functions of actomyosin contractility in “neuronal compartments”. However, the paper is not organized to focus on compartments. It is suggested that the organization of the manuscript be revised to present sections that cover the main relevant issues: AIS, axon shaft, dendritic spines. As presented, the text shifts from one to the other.

The discussion of calpains, as presented, would also benefit from reorganization and placement in relation to text involving actomyosin. Lines 157-186 deal extensively with calpains although the relationship to actomyosin is not made in this section.

Specific comments:

line 37 "...where it interconnects microtubule bundles with actin-rich filopodia". This statement about NMII could be misinterpreted to mean that NMII physically connects microtubules and actin filaments in filopodia. It does not. The referenced papers (1,2) provide evidence that NMII restricts the advance of microtubules into the peripheral domain of growth cones and promotes the bundling of microtubules during the process of consolidation. This would also be an appropriate location to reference other papers from P Forscher on the issues of NMII and microtubule distribution in growth cones.

Line 42-43. The leading sentence of this section applies mostly to central nervous system neurons and not sets of peripheral neurons that instead generate the axon without multiple precursor processes. It would be cautious to note this for readers.

line 47 "in dependence of microtubule bundling" is the intent to say "dependent on microtubule bundling"? Also, it might be useful to at least briefly mention the roles of microtubule polymerization and transport, which are prerequisites to their bundling in situ as the axon grows.

Line 78. It is unclear to this reader how "Neuronal periodic membrane cytoskeleton" translates into the abbreviation given throughout the text of "MPS". Perhaps Periodic Membrane Cytoskeleton as PMC?

Line 100. "Spectrins are required for MPS assembly and the stability of axons [29]". Reference 29 does not address the role of spectrins in the assembly of the MPS. Indeed, reference 29 is from a 2007 publication while the MPS was first described in 2013. It would also be useful to provide a clearer exposition of the findings of 29 in terms of the what is meant  by the "stability" of axons. 29 reports axon fragmentation in response to naturally occurring movements in c. elegans mutants of spectrin. Similarly, reference 30 could use additional description of its findings.

Line 111. The Axon Initial Segment is characterized as a compartment "that separates the axonal and dendritic domains of neuronal cells". It separates the axon from the cell body/soma. Perhaps somatodendritic domain could be stated.

lines 100-123. "AIS is required for the maintenance of neuronal polarity". The authors argue that because hypoxia causes AIS breakdown and the loss of polarity the AIS is required for maintenance of polarity. Hypoxia impacts the cell as a whole and the interpretation provided is considered confounded. The claim would only be supported if specific studies can be referenced where the AIS is specifically disassembled resulting in the loss of polarity. The authors state on lines 123-126 that AnkG depletion similarly results in loss of polarity, but do not provide a reference. Thus, it is suggested that the section start with the AnkG study (and a reference be provided) and follow with the indirect observations on hypoxia.

Line 128-129. Define "prolonged signaling". This could be interpreted in a number of ways, but I gather the authors refer to patterns of action potentials.

The last sentence of  section 3 states a role for myosin II at the AIS, the main theme of the review. However, it is not until line 187 that the issue is addressed leaving the reader "hanging".

Line 145. Differential resistance to drug induced depolymerization is not usually considered to be reflective of "diversity on structural organization" of actin filaments, but rather their rates of turnover. Capped filaments, such as those in axonal rings that are bound by the barbed end capping protein alpha-adducin are expected to be undergoing less turn over than non capped filaments as most turnover is through filament end polymerization/depolymerization. It is suggested the statement be revised to more accurately interpret the difference in the response to depolymerizing drug treatments as reflective of differences in filament turnover.

lines 147-156. The paragraph discusses the physical properties of the submembranous cytoskeleton and ends with a statement that the role of actomysoin is now a focus of research into this issue. However, as previously noted for another statement about the role of myosin, the reader is left hanging as the next paragraph does not deal with myosin II in any way.

Line 159 "... in dependence of ERK signaling" the meaning is unclear. Is the intent to say that Erk activity drives calpain-2 activity?

Line 160. "Calpain-1 and -2 are major cellular proteases that are critical for proper neuronal branching and dendritic spine complexity [41]." Reference [41] contains no data addressing neuronal branching or spine complexity and the role of calpains. Similarly, how reference 41 supports the statement about calcium on lines 161-163 is also not clear.

line 167 "Functional cross-talk between clustered molecules triggers the activation of Erk..." please rewrite and define that receptors are being discussed and not "molecules", also strive to use more conventional language as used in the receptor signal transduction field.

Line 170 "Interestingly, few hours later the MPS can be fully recovered [40]." Recovered from what? Please specify.

Line 176. A ( is opened but never closed with a ). Also, the sentence needs consideration as it is a list of more than a sentence.

How lines 178 "Accordingly.... disease [48,49]" mentioning cataracts and cancer relate to the focus of the review is unclear.

Although myosin is at the core of this review, it is not properly introduced until lines 187-211. The organizational flow of the paper is not considered ideal. Myosin in introduced after a section on calcium and calpains, which does not have a clear relationship to the focus of the review as stated in the title and abstract which is actomyosin.

line 217. "both types of the AIS plasticity". It is not clear what two types are being referred to as the prior text in the paragraph seems to describe only AIS restructuring in response to depolarization. Clarification is warranted by specifically stating what two types the authors are referring to.

In discussing reference [4] (e.g., lines 219 on) the authors must be cautious to note that diphosphorylated MLC is being discussed. The issue of di relative to mono phosphorylation of MLC must also be introduced to provide context discussing the functional differences between mono and di phosphorylation. As written, a reader may get the impression that MLC is only phosphorylated (in any form) at the AIS, which is very misleading.

line 239 seems to have an inadvertent end of paragraph.

line 241 "of a large cargos": (1) single plural grammatical mismatch, (2) cargoes not cargos.

Lines 241-243 make a claim that intracellular organelles have diameters larger than the 1 micron of small axons and references [56] for autophagosomes, mitochondria, endosomes and lysosomes. The term diameter is only used once in [56] "The diameter of a typical autophagosome is approximately 500 nm". Overall, the statement is not correct as these organelles generally have diameters equal to or smaller than the axon, even for the small diameter axons although that can approach equality in diameter. However, for the smallest of axons the claim does generally apply but reference [39] would be best suited here as in their introduction these authors actually cover the range of organelles discussed herein.

lines 244-256 the work of "Wang and colleagues" and then "the Sousa laboratory" is discussed and multiple lines of text are presented before a reference is provided. It is suggested that referencing occur at all relevant places where statements regarding data are presented. Also, stylistically the focus on investigators for a few select references seems unwarranted and it is suggested that the style of presentation maintain the focus on the data.

In discussion [3] the authors ought to clearly present which form on MLC phosphorylation is being addressed, as noted in the previous comment about di vrs mono phosphorylation.

line 272 "(Xu et al., 2013)" deviates from standard referencing throughout. There are additional examples of this in the Perspectives section, a peculiar oversight.

Line 271-275. This leading sentence requires some rewriting as it runs on and has grammatical issues (e.g., "... as well as presents across different ...).

line 275 "C. elegance" likely an autocorrect not corrected by the authors. But also, "In the study where C. elegance was used..." the statement is not ideal as "the study" in question was not discussed since the early part of the paper and the model system was not at that point stated.

Line 276, sentence starts with "And".

Line 290 "This offers an explanation how narrow" revise to "This offers an explanation for how narrow"

Line 295 "possibilities to connect" did the authors intend “to consider”?

Overall, the Perspectives section is poorly written and requires revision with an emphasis on grammar and word choice.

Also, there is no section 5 between 4 and 6.

In discussing the relationship of myosin II to actin filaments in rings the authors ought to further consider the geometry of the head  domain binding to the polarity of filaments and how this then relates to the organization of the rest of the heavy chain relative to the filaments.

Box 1. Line 392. Microtubules have plus ends, actin filaments have barbed ends. Although in the neuroscience field these two terms have started to be used as if equivalent, they are not and barbed ends should be discussed for actin filaments.

Box 1. The description provided refers to filaments arrayed with opposite polarity as in a sacromere but it is not specifically stated.

Box 1. Line 413. It is unclear why the authors consider only beta actin and not gamma to be of relevance in their presentation as gamma is also expressed by neurons.

Author Response

(The authors gave the same response as above.)

Round 2

Reviewer 2 Report

The authors adequately attended to the suggested revisions.

"lines 244-256 the work of "Wang and colleagues" and then "the Sousa laboratory" is discussed and
multiple lines of text are presented before a reference is provided. It is suggested that referencing
occur at all relevant places where statements regarding data are presented. Also, stylistically the focus
on investigators for a few select references seems unwarranted and it is suggested that the style of
presentation maintain the focus on the data.

Reply: We thank the reviewer for the generous description of his personal stylistic preferences and
added a few extra citations of the Wang et al paper to the paragraph describing the work therein."

I would like to emphasize that no generosity was involved in the reviewer's comment. Rather, the comment addressed a deviation from style of presentation used throughout the manuscript without an obvious need for the deviation.